# InComeS: Integrating Compression and Selection Mechanisms into LLMs for Efficient Model Editing

## Abstract

Although existing model editing methods perform well in recalling exact edit facts, they often struggle in complex scenarios that require deeper semantic understanding rather than mere knowledge regurgitation. Leveraging the strong contextual reasoning abilities of large language models (LLMs), in-context learning (ICL) becomes a promising editing method by comprehending edit information through context encoding. However, this method is constrained by the limited context window of LLMs, leading to degraded performance and efficiency as the number of edits increases. To overcome this limitation, we propose *InComeS*, a flexible framework that enhances LLMs' ability to process editing contexts through explicit compression and selection mechanisms. Specifically, InComeS compresses each editing context into the key-value (KV) cache of a special gist token, enabling efficient handling of multiple edits without being restricted by the model's context window. Furthermore, specialized cross-attention modules are added to dynamically select the most relevant information from the gist pools, enabling adaptive and effective utilization of edit information. We conduct experiments on diverse model editing benchmarks with various editing formats, and the results demonstrate the effectiveness and efficiency of our method.

## 1 Introduction

Model editing, also known as knowledge editing, has seen rapid progress in recent years (Fang et al., 2024; Li et al., 2024; Wang et al., 2024; Zhang et al., 2024a). Its primary goal is to precisely integrate updated knowledge into a model, enabling targeted behavioral modifications while maintaining performance on unrelated tasks. Existing techniques have demonstrated strong performance in accurately recalling edited facts (Yao et al., 2023; Zhang et al., 2024a; 2025). However, they often struggle in more complex editing scenarios, such as multi-hop editing composition (Zhong et al., 2023b; Zhang et al., 2025), natural language editing (Akyürek et al., 2023), and editing tasks that require reasoning and generalization (Cohen et al., 2024; Zhang et al., 2024a). Moreover, recent studies (Zhang et al., 2025) show that previous editing methods are prone to overfitting: they may assign excessively high probabilities to edited targets, which can distort the model's responses to more complex or nuanced queries.

Leveraging the in-context learning (ICL) abilities of large language models (LLMs) provides a promising direction for addressing these problems. As LLMs continue to grow in size and capability, their ability to understand and utilize contextual information continues to improve. By incorporating all the editing information into the prefix contexts, ICL enables a simple, powerful, and flexible approach for employing updated knowledge in complex scenarios. However, this approach faces significant challenges as the number of edits increases. First, the finite context window restricts the maximum number of edits that can be included, and the computational cost of self-attention over long contexts leads to a sharp decline in *efficiency*. Moreover, the effectiveness of ICL is constrained by the model's ability to process extended contexts, and the retrieval *accuracy* of the most relevant editing information also tends to decrease as the editing context grows.

To address these challenges, we introduce *InComeS* (**In**tegrating **Comp**ression and **S**election Mechanisms), a novel framework for efficient and scalable model editing. InComeS adopts context compression techniques to condense the representation of each edit into the KV cache of special gist to-

kens, which can be cached and reused for computational efficiency. While gisting (Mu et al., 2023) was originally developed to compress single-input prompts, we extend this approach to handle multiple edits by further introducing a specialized selection mechanism. Specifically, we augment the model with cross-attention modules that allow each input token to attend to the compressed gist representations of edits, enabling fine-grained and adaptive selection of the most relevant information. Since each edit is compressed in parallel, our framework overcomes the limitations imposed by the context window, and the specialized selection modules can be learned to enhance retrieval accuracy.

We conduct experiments across a range of complex model editing settings, including multi-hop editing, natural language editing, and tasks requiring complicated reasoning. Experimental results demonstrate that InComeS outperforms existing editing methods, effectively handling diverse editing scenarios while offering efficiency gains.

## 2 PRELIMINARY

Model editing (Yao et al., 2023; Mitchell et al., 2022a) aims to adjust a base model $\psi$ to a post-edited model $\psi'$ according to a set of editing information $\mathcal{T} = \{t_1, \ldots, t_n\}$:

$$\psi' = \text{Edit}(\psi, \{t_1, \ldots, t_n\}) \tag{1}$$

Here, "Edit" indicates the model editing method, while $\{t_1, \ldots, t_n\}$ represents the knowledge pieces to be integrated. A typical example of editing information is query-label pair $t = (x, y)$, where the goal is for the edited model to produce $y$ in response to input $x$, even if the original model does not:

$$\psi(x) \neq y, \quad \psi'(x) = y \tag{2}$$

When the editing set contains only a single piece of information ($|\mathcal{T}| = 1$), this is known as single-instance editing. In contrast, batch editing refers to the scenario where multiple pieces of knowledge are updated simultaneously ($|\mathcal{T}| > 1$). Batch editing is particularly practical in real-world applications, where simultaneously updating several edits is often required. In these scenarios, it will be more efficient to integrate them into the model in a single operation.

In practice, editing information can take various forms beyond simple query-label pairs. For instance, multiple related edits can be combined to enable multihop editing, or updated knowledge may be provided as a paragraph of natural language text. In such scenarios, many traditional editing methods may struggle to produce the desired outcomes, since they are not designed to handle these diverse types of edit information. In contrast, in-context learning (ICL) approaches, where editing information is simply concatenated as contextual prefixes, offer a straightforward yet powerful solution:

$$\text{Edit}_{\text{ICL}}(\psi, \{t_1, \ldots, t_n\})(x) = \psi(t_1, \ldots, t_n, x) \tag{3}$$

By leveraging the LLM's ability to understand and reason over context, ICL can naturally accommodate a wide range of editing scenarios. Nevertheless, ICL is constrained by the context window of LLMs, and its accuracy and efficiency tend to decline when processing larger batches of edits.

## 3 METHOD

In this work, we aim to enhance the ICL-based editing approach to better understand multiple edits and accurately extract relevant information from the edit batch. Given a batch of editing information set $\mathcal{T} = \{t_1, \ldots, t_n\}$, an input query $x$, and the subset of its related[1] edits $\{t_i | i \in \mathfrak{R}(x)\}$, we hope that our model can answer the query as effectively as a vanilla LM provided only with the relevant edits (ignoring the irrelevant editing information):

$$\psi' = \text{Edit}(\psi, \{t_1, \ldots, t_n\}) \approx \text{Edit}(\psi, \{t_i | i \in \mathfrak{R}(x)\}) \tag{4}$$

To enable accurate and efficient batch editing, we propose **_InComeS_** an ICL-based approach that integrates both compression and selection mechanisms into the LMs. First, we adopt gist-based edit compression, condensing each editing information into the KV cache representations of one special (gist) token. Furthermore, we introduce parallel-context cross-attention modules that allow ordinary tokens to attend to these compressed gist representations. These modules serve as soft selectors to dynamically identify the most relevant information for the current input. This strategy can effectively mitigate the limitations imposed by context window sizes and enhance the model's ability to precisely capture editing information.

---

[1]We define related edits as those editing pieces that the model should reference when answering the query.

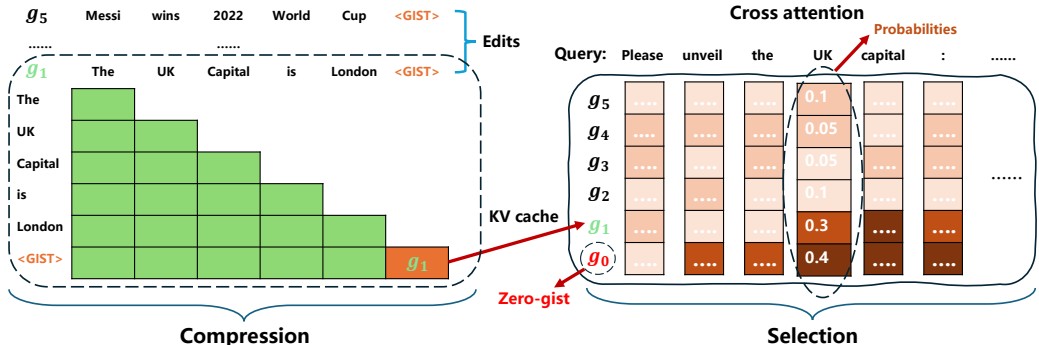

Figure 1: An overview of *InComeS*. At the compression stage, each edit is individually condensed into KV cache representations of a gist token. These representations are integrated into the model via selection through the cross-attention modules. A special zero-gist token is included alongside the cached gists from actual edits, allowing the model to have the option to "select nothing." Note that the compression and integration steps are performed separately, but both use the same underlying model.

## 3.1 EDIT COMPRESSION

We adopt the concept of gisting (Mu et al., 2023), which is originally developed to compress input prompts into the representations of an extra, specially inserted token (the gist token). The condensed gist activation serves the same function as the original prompt and can be cached for later reuse, thereby improving computational and memory efficiency. While the original work primarily focuses on instruction tuning, we extend this idea to edit compression.

For each editing information piece $t_i$, represented as a sequence of tokens $t_i^0, t_i^1, \ldots, t_i^n$, we append a special gist token $t_g$ to the end of the sequence and feed it into the LM. After encoding, we discard the original edit tokens and retain only the gist's representations (KV caches) for each edit. Notably, each edit context is encoded independently, allowing us to efficiently handle an arbitrary number of edits. This approach is highly flexible and accommodates edits of varying lengths and formats.

After edit compression, the edit information $t_1, t_2, \ldots, t_n$ is converted into their corresponding gist KV representations[2] $(gK_1, gV_1), (gK_2, gV_2), \ldots, (gK_n, gV_n)$. Importantly, we use the same LM targeted for editing to encode and compress the edit information, ensuring that the subsequent information selection process is seamless and well-aligned with the model's internal representations.

## 3.2 EDIT SELECTION

After compressing the edit contexts, we obtain a pool of gist representations for the batch of edits. To integrate this information into the model, we introduce additional cross-attention modules that enable input tokens to attend to the edit representations. Since these representations are stored as KV caches, we leverage a similar attention mechanism to incorporate the edit information. Formally, given a token's query state $q$, the cross-attention is computed as:

$$o_{cross} = attention(q, \{gK_0, gK_1, gK_2, \ldots, gK_n\}, \{gV_0, gV_1, gV_2, \ldots, gV_n\}) \quad (5)$$

We finally add the cross-attention outputs to the self-attention outputs for information aggregation.

Since tokens are not required to always attend to the edit information, we further introduce a zero-gist ($g_0$ in Figure 1) to allow the model to attend to "nothing" when appropriate. For the zero-gist, we use learnable parameters for the key vectors $gK_0$ and assign fixed zero vectors to the value $gV_0$. This design allows the model to flexibly select relevant information as needed during sequence prediction.

## 3.3 TRAINING

Since vanilla LMs lack explicit mechanisms for context compression and selection, we perform continued training (Figure 2) to enhance pre-trained LMs with these capabilities. Our main goal

---

[2]For brevity, we present the representations and operations for a single layer.

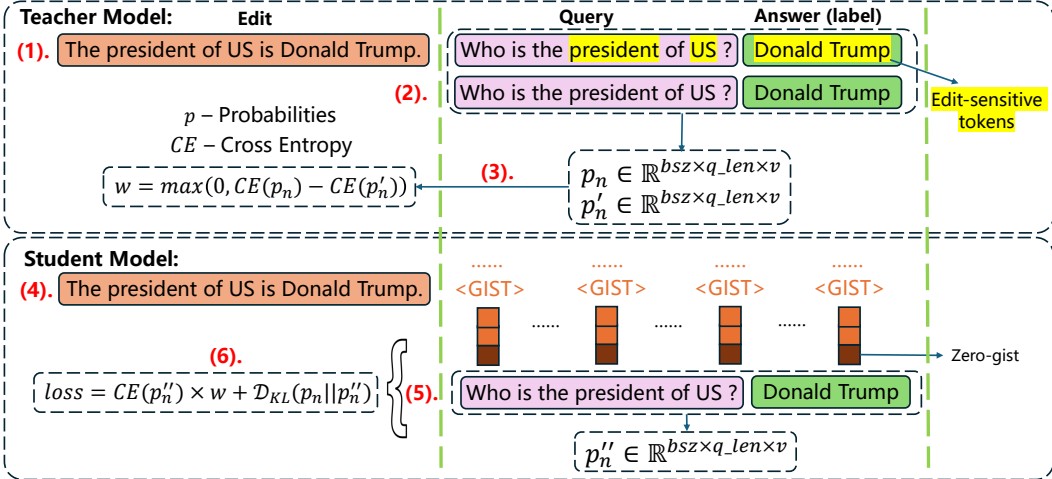

Figure 2: An overview of training *InComeS*. The teacher model performs two forward passes: one with edit-contextualized input (1) and one with uncontextualized input (2). The cross-entropy between the outputs of (1) and (2) is used to compute a customized weight (3). The student model then compresses the edit information into KV representations using gist tokens (4). These KV caches are used to supply edit-relevant information to the query tokens (5). The final loss is computed as the sum of weighted cross entropy and KL divergence (6).

is to ensure that the compressed gist representations serve as effective substitutes for the original editing information. To achieve this, it is essential to distinguish between edit-sensitive tokens, whose losses change significantly when editing context is given, and edit-insensitive tokens, which can be predicted accurately from local context alone and do not depend on edit information. This distinction is captured by employing a customized token weighting scheme:

$$w_{x_i} = \max(0, \ CE(x_i|x_0, \dots, x_{i-1}) - CE(x_i|\{t_i|i \in \mathfrak{R}(x)\}, x_0, \dots, x_{i-1})) \quad (6)$$

where the $CE$ is the cross entropy and $\mathfrak{R}(x)$ represents the subset of related edits. Here, the token weight is the difference between the edit-conditioned and edit-unconditioned losses. This scheme increases the weights of edit-sensitive tokens to encourage the model to learn to retrieve information from the compressed edits. The loss differences are calculated with a teacher model, which is the original, unedited version of the target LM.

In addition to token reweighting, we also adopt knowledge distillation (Hinton et al., 2015) to transfer the teacher model's knowledge about the edit information into the target model. Specifically, we apply the KL divergence to align the output distributions of the gist-contextualized student model with those of the edit-contextualized teacher model:

$$KL_{x_i} = D_{KL}(p_T(x_i|\{t_i|i \in \mathfrak{R}(x)\}, x_0, \dots, x_{i-1}) \ || \ p_S(x_i|\{g_1, \dots, g_n\}, x_0, \dots, x_{i-1})) \quad (7)$$

$$loss_{x_i} = w_{x_i} \cdot CE(x_i) + KL_{x_i} \quad (8)$$

Here, $g_1, \dots, g_n$ denote the cached gists for all the edits. We apply the token reweighting only to the vanilla cross-entropy term in our final loss, since we found that it would degrade effective learning of "attend-to-nothing" behavior if combined with the KL part. The explicit training details are provided in Appendix C.

## 4 EXPERIMENTS

### 4.1 EXPERIMENT SETTING

**Datasets & Evaluation Metrics**  To verify the effectiveness of our method in complex editing scenarios, we conduct experiments on five popular datasets in model editing: the dataset for multi-hop editing MQuAKE (Zhong et al., 2023b), the natural language editing dataset DUNE (Akyürek et al., 2023), the extended version of ZsRE (Yao et al., 2023; Zhang et al., 2024a), which adds a portability test set to the original ZsRE (Levy et al., 2017), and the dataset containing ripple effect

| Method | Model | Single Editing | | | Batch Editing | | |
|---|---|---|---|---|---|---|---|
| | | 2-edits | 3-edits | 4-edits | 2-edits | 3-edits | 4-edits |
| Base | | 41.79 | 43.51 | 31.58 | 41.79 | 43.51 | 31.58 |
| FT-M | | 55.32 | 56.59 | 52.22 | 51.89 | 50.15 | 44.1 |
| LoRA (Hu et al., 2022) | | 67.63 | 68.84 | 60.21 | 50.29 | 47.88 | 47.15 |
| ROME (Meng et al., 2022) | | 1.97 | 7.77 | 0.55 | - | - | - |
| R-ROME (Gupta et al., 2024a) | | 2.72 | 4.29 | 8.53 | - | - | - |
| MEMIT (Meng et al., 2023) | Llama-3.2-1B | 40.21 | 39.54 | 22.34 | 29.71 | 28.87 | 15.7 |
| EMMET (Gupta et al., 2024b) | | 5.4 | 6.61 | 4.74 | 11.49 | 17.16 | 15.89 |
| GRACE (Hartvigsen et al., 2022) | | 8.45 | 7.13 | 5.24 | 2.03 | 2.68 | 2.06 |
| SERAC (Mitchell et al., 2022b) | | 41.84 | 43.51 | 31.58 | 41.84 | 43.45 | 31.74 |
| MEND (Mitchell et al., 2022a) | | 39.88 | 35.23 | 33.31 | 35.55 | 30.29 | 29.89 |
| ICL | | 59.23 | 59.00 | 51.63 | 50.06 | 49.86 | 42.37 |
| **InComeS** | | **71.19**$_{20.19\%\uparrow}$ | **72.17**$_{22.32\%\uparrow}$ | **72.62**$_{40.65\%\uparrow}$ | **53.93**$_{7.73\%\uparrow}$ | **52.79**$_{5.88\%\uparrow}$ | **52.73**$_{24.45\%\uparrow}$ |
| Base | | 44.08 | 44.14 | 30.62 | 44.08 | 44.14 | 30.62 |
| FT-M | | 69.89 | 74.88 | 74.96 | 50.31 | 49.04 | 50.39 |
| LoRA (Hu et al., 2022) | | 36.95 | 34.28 | 29.02 | 16.41 | 24.22 | 23.12 |
| ROME (Meng et al., 2022) | | 9.67 | 7.33 | 8.93 | - | - | - |
| R-ROME (Gupta et al., 2024a) | | 10.73 | 6.68 | 3.62 | - | - | - |
| MEMIT (Meng et al., 2023) | Qwen2.5-7B | 44.14 | 46.05 | 32.15 | 43.94 | 46.10 | 30.55 |
| EMMET (Gupta et al., 2024b) | | 26.10 | 38.38 | 26.36 | 40.83 | 45.02 | 33.89 |
| GRACE (Hartvigsen et al., 2022) | | 15.78 | 13.45 | 13.23 | 5.43 | 7.88 | 4.23 |
| SERAC (Mitchell et al., 2022b) | | 55.56 | 59.23 | 53.67 | 42.34 | 40.33 | 39.39 |
| MEND (Mitchell et al., 2022a) | | 34.23 | 45.34 | 30.25 | 39.88 | 35.45 | 34.21 |
| ICL | | **69.76** | **76.91** | 74.54 | 53.53 | 50.54 | 44.77 |
| **InComeS** | | 66.46$_{4.73\%\downarrow}$ | 71.24$_{7.37\%\downarrow}$ | **76.54**$_{2.68\%\uparrow}$ | **55.13**$_{3.00\%\uparrow}$ | **53.48**$_{5.82\%\uparrow}$ | **47.91**$_{7.01\%\uparrow}$ |

Table 1: Results on MQuAKE (Zhong et al., 2023b). The difference between InComeS and ICL is marked.

samples WikiData$_{counterfact}$ (Cohen et al., 2024; Zhang et al., 2024a). We report edit success rate and portability for the extend-ZsRE and WikiData$_{counterfact}$, the results for 2, 3, and 4 edits for MQuAKE (Zhong et al., 2023b) and new information, scientific reasoning, and debiasing for DUNE (Akyürek et al., 2023). More details about the datasets and evaluation metrics can be found in the Appendix D.1 and Appendix D.2, respectively.

**Baselines** For baselines, we select representative methods demonstrated to be powerful in relevant surveys (Yao et al., 2023; Zhang et al., 2024a). For methods that directly edit the model weights, we include ROME (Meng et al., 2022), R-ROME (Gupta et al., 2024a), and MEMIT (Meng et al., 2023); for methods that adopt explicit external memory, we include SERAC (Mitchell et al., 2022b), and IKE (Zheng et al., 2023); for methods that train additional meta-model or use implicit external memory (stores activations or nerons, etc), we adopt MEND (Mitchell et al., 2022a), GRACE (Hartvigsen et al., 2022) and KN (Dai et al., 2022). We also include the traditional but powerful method, like fine-tuning, LoRA (Hu et al., 2022), and ICL, which directly concatenates all the edits as the prefix context. While some similarities exist between our method and RAG, they vary considerably in problem setting and methodology. A detailed analysis is given in Appendix E.1. We choose two representative open-source models for evaluation: Llama-3.2-1B [3] and Qwen2.5-7B (Yang et al., 2024). Unless otherwise specified, we adopt an edit batch size of 100 for batch editing. More details on the baseline implementation can be found in the Appendix D.3.

### 4.2 MAIN RESULTS

**Multi-hop edits** We test our method on MQuAKE for the multiple-hop edit scenario, where the models are required to check multiple edits to answer each query. Because of this requirement, we mainly compare with methods designed to support batch or sequential editing. Table 1 presents our main results, which demonstrate the effectiveness of InComeS in both single-editing and batch-editing scenarios. In addition, InComeS surpasses ICL in all metrics except single 2-edits and 3-edits for Qwen2.5-7B, which shows that our method can effectively select relevant information from the editing contexts. Interestingly, single-edit specialized methods (such as ROME) collapse even in the single multi-hop query setting, revealing their incapability to handle complex editing scenarios.

**Natural language edits** One of our method's advantages is its flexibility in handling a variety of editing contexts with different formats. Unlike many traditional editing methods like ROME (Meng et al., 2022) and MEMIT (Meng et al., 2023), which require the input to follow the triplet-like fact statement format, InComeS can take edits in free-text forms without explicitly labeled subjects and objects. To verify our method's capability for such scenarios, we adopt the DUNE dataset, which

---

[3] https://huggingface.co/meta-llama/Llama-3.2-1B

| Method | Model | Single Editing | | | Batch Editing | | |
|---|---|---|---|---|---|---|---|
| | | New info | Scientific R. | Debiasing | New info | Scientific R. | Debiasing |
| Base | | 56.85 | 55.87 | 32.73 | 56.85 | 55.87 | 32.73 |
| FT-M | | 57.43 | 53.34 | 35.73 | 57.07 | 53.45 | 33.43 |
| LoRA | Llama-3.2-1B | 53.65 | 50.87 | 36.73 | 56.77 | 54.66 | 35.83 |
| SERAC (Mitchell et al., 2022b) | | 52.67 | 48.96 | 46.73 | 50.76 | 47.32 | 36.52 |
| ICL | | 58.67 | 55.84 | **56.65** | 56.94 | 55.68 | 39.46 |
| **InComeS** | | **60.00**$_{2.27\%\uparrow}$ | **58.17**$_{4.17\%\uparrow}$ | 54.61$_{3.60\%\downarrow}$ | 57.76$_{1.44\%\uparrow}$ | **56.46**$_{1.40\%\uparrow}$ | **46.14**$_{16.93\%\uparrow}$ |
| Base | | 63.44 | 66.03 | 36.95 | 63.44 | 66.03 | 36.95 |
| FT-M | | 64.83 | 67.77 | 48.73 | 64.27 | 64.56 | 41.51 |
| LoRA | Qwen2.5-7B | 63.85 | 63.22 | 39.56 | 62.58 | 65.33 | 43.73 |
| SERAC (Mitchell et al., 2022b) | | 64.45 | 63.57 | 33.38 | 56.78 | 58.97 | 31.24 |
| ICL | | 65.81 | 65.34 | 35.25 | **66.29** | 66.24 | 40.73 |
| **InComeS** | | **66.83**$_{1.55\%\uparrow}$ | **68.02**$_{4.10\%\uparrow}$ | **62.59**$_{77.56\%\uparrow}$ | 65.61$_{1.03\%\downarrow}$ | **67.82**$_{2.39\%\uparrow}$ | **56.69**$_{39.18\%\uparrow}$ |

Table 2: Results on DUNE (Akyürek et al., 2023). The difference between InComeS and ICL is marked.

includes *natural-language form edits*, and the results are shown in Table 2. Following the original paper of DUNE (Akyürek et al., 2023), we include fine-tuning, LoRA, SERAC (Mitchell et al., 2022b), and ICL as our baselines. The result confirms our method's capability to handle natural language edits. Interestingly, the raw model itself is a strong baseline in the batch editing scenario, which may demonstrate the fast-evolving model capabilities over the years.

**Evaluation on portability** We further evaluate our method on two popular editing datasets that require reasoning abilities: $Wiki_{counterfact}$ (Cohen et al., 2024; Zhang et al., 2024a) and the extended ZsRE (Yao et al., 2023; Zhang et al., 2024a). Table 3 shows the results.[4] Our primary focus is on portability, as it serves as the most representative metric for assessing a model's comprehensive understanding of the editing information. Overall, our method achieves performance comparable to ICL and consistently outperforms other baselines. As expected, traditional editing methods such as fine-tuning and ROME exhibit high edit success rates; however, their portability scores generally lag behind the top-performing methods, highlighting a common limitation of these approaches. In contrast, ICL-based approaches that leverage the in-context learning capabilities of LLMs demonstrate superior performance in complex editing scenarios that require reasoning, owing to the enhanced context understanding of LLMs.

**Scaling up contexts** We further provide a scaling-up analysis to illustrate our method's ability to generalize to larger numbers of edits, which is the main motivation of our modification over the ICL baseline. For this analysis, we use the COUNTERFACT dataset (Meng et al., 2023), as it provides a sufficient number of editing instances. We vary the number of edits from 100 to 1000, resulting in total token counts ranging from approximately 1.2K to 12K. The results are shown in Figure 3, which shows that InComeS consistently outperforms ICL, though the base models have already been pretrained over long contexts (Yang et al., 2024). This finding suggests that the vanilla attention mechanism alone is insufficient to effectively comprehend and precisely select the required information from the context in complex editing scenarios. In contrast, our method demonstrates greater potential for handling large-scale edits through the unified compression and selection mechanism.

### 4.3 ABLATION STUDY & ANALYSIS

**Inclusion of zero gist** The motivation of including the zero-gist mechanism is to ensure that context-independent tokens can bypass the influence of the edit contexts. To assess the impact of zero-gist, we train a model without this mechanism and evaluate it on DUNE (Akyürek et al., 2023) and MQuAKE (Zhong et al., 2023b) (see the "w/o zero-gist" line in Table 4). The results show a notable performance drop, particularly in multi-hop scenarios, suggesting that the cross-attention calculations may sometimes interfere with ordinary generation and our zero-gist strategy can mitigate this issue by allowing tokens to "attend to nothing".

**Full model vs. Half model** We present the reason for our decision to use the KV cache from the second half of the model layers. To investigate this, we train a model using the KV cache from all layers and evaluate it on 1000 instances from ZsRE (Levy et al., 2017). We record the probabilities allocated to the zero-gist in the cross-attention modules, as shown in Fig. 4b. The

---

[4]We only present methods deemed representative and powerful on these two datasets, detailed version can be found in Table 7.

| Method | Model | WikiData$_{counterfact}$ | | ZsRE-extended | |
|---|---|---|---|---|---|
| | | Edit Success | Portability | Edit Success | Portability |
| Base | | 21.28 / 21.28 | 19.73 / 19.73 | 30.06 / 30.06 | 40.17 / 40.17 |
| FT-M | | **97.02** / **94.58** | 53.43 / 47.51 | **99.81** / **95.94** | 62.80 / **54.84** |
| LoRA (Hu et al., 2022) | | **98.91** / 82.61 | 52.87 / 43.84 | **99.86** / **93.18** | 57.43 / 44.85 |
| ROME (Meng et al., 2022) | | 94.33 / - | 40.44 / - | 95.41 / - | 46.04 / - |
| MEMIT (Meng et al., 2023) | | - / 66.94 | - / 23.51 | - / 58.79 | - / 25.68 |
| MEND (Mitchell et al., 2022a) | Llama-3.2-1B | - / 26.66 | - / 21.06 | - / 43.33 | - / 30.77 |
| GRACE (Hartvigsen et al., 2022) | | 33.27 / 25.06 | 14.33 / 10.51 | 32.00 / 24.44 | 12.73 / 10.91 |
| IKE (Zheng et al., 2023) | | 61.70 / - | 45.55 / - | 59.15 / - | 57.39 / - |
| SERAC (Mitchell et al., 2022b) | | 89.56 / 78.32 | 60.56 / 40.45 | 92.69 / 89.61 | **66.59** / 51.61 |
| ICL | | 93.31 / **82.95** | **65.81** / **49.75** | 68.86 / 60.84 | 62.19 / **55.58** |
| **InComeS** | | 91.16 / 76.81 | **65.15** / 45.66 | 97.22 / 87.09 | **70.70** / 52.23 |
| Base | | 22.35 / 22.35 | 21.46 / 21.46 | 36.21 / 36.21 | 43.86 / 43.86 |
| FT-M | | **98.93** / **90.18** | 49.39 / 43.13 | **99.51** / **92.60** | 50.04 / 46.41 |
| LoRA (Hu et al., 2022) | | 77.31 / 72.22 | 37.04 / 31.65 | 86.88 / 77.78 | 28.61 / 24.13 |
| ROME (Meng et al., 2022) | | 92.69 / - | 40.25 / - | 97.86 / - | 50.43 / - |
| MEMIT (Meng et al., 2023) | | - / **91.16** | - / 39.85 | - / **93.28** | - / 49.97 |
| MEND (Mitchell et al., 2022a) | Qwen2.5-7B | - / 35.13 | - / 15.29 | - / 50.91 | - / 38.83 |
| GRACE (Hartvigsen et al., 2022) | | 31.34 / 33.77 | 25.60 / 18.55 | 33.27 / 26.79 | 14.35 / 11.25 |
| IKE (Zheng et al., 2023) | | **96.40** / - | **75.33** / - | **99.75** / - | 83.17 / - |
| SERAC (Mitchell et al., 2022b) | | 91.79 / 80.68 | 51.12 / 41.26 | 91.12 / 82.56 | 62.41 / 52.63 |
| ICL | | 90.24 / 85.28 | **66.99** / **51.66** | 71.75 / 71.57 | **66.10** / **64.57** |
| **InComeS** | | 90.96 / 71.44 | 66.69 / **47.93** | 97.95 / 91.29 | **75.63** / **61.22** |

Table 3: Results for $WikiData_{counterfact}$ (Cohen et al., 2024; Zhang et al., 2024a) and ZsRE-extended (Yao et al., 2023; Zhang et al., 2024a). The data format of each cell is in "single/batch editing results". "-" means the methods are not designed for the corresponding settings. The best two statistics are marked. Full results can be found at Table 7.

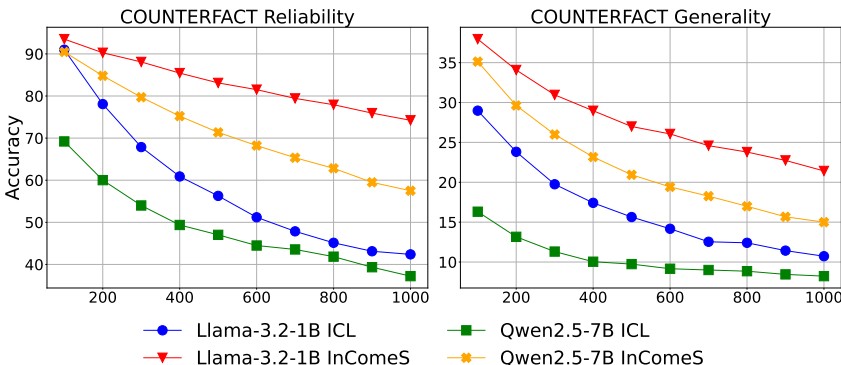

Figure 3: Scaling-up analysis. We compare InComeS and ICL by varying the number of edits, as indicated on the $x$-axis.

result shows that the zero-gist probabilities in layers 7-15 are generally lower than those in layers 1-6, and there is a notable drop in the zero-gist probability at layer 7. This suggests that, even when trained to use the full-model KV cache, the model mainly relies on information from deeper layers, since higher probabilities of the zero-gists indicate lower utilization of the actual edit contexts. A possible explanation is that more information is accumulated in the deeper layers, which aids both compression and selection processes. To verify our analysis, we also test the full layer trained model (see the "w/ full model" line in Table 4) under a batch editing setting, and observe a general decrease across all metrics. Additionally, restricting the KV cache to only the second half of the model could provide efficiency benefits with lower memory and computation costs.

**Applying loss on queries** By convention, instruction tuning only takes into account the loss for labels, excluding queries (Fig. 2). In this section, we show that merely applying a loss on labels is not enough in our case. We train a model without the loss of queries and present its results in the Table 4 (the line of "w/o loss on query"). The absence of query loss results in a sharp decrease for multi-hop editing, suggesting that training on query tokens may improve the model's capability of combining information retrieval and reasoning.

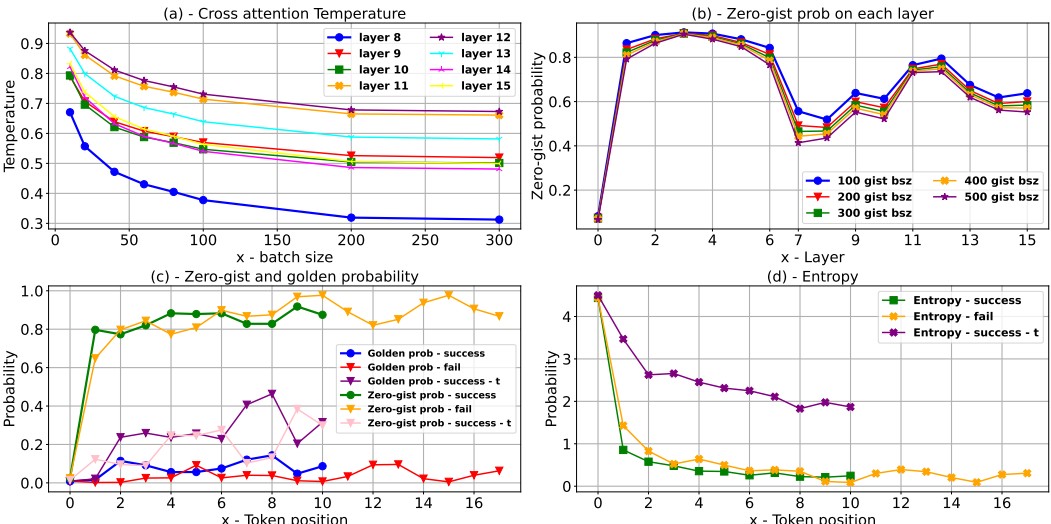

Figure 4: Ablation and Analysis. (a) Experiments to investigate the desired temperature for cross-attention. (b) Investigation on the informativeness of the layers. (c) and (d) Study to reveal the selection pattern of the query tokens.

| Method | Model | DUNE | | | MQuAKE | | |
|---|---|---|---|---|---|---|---|
| | | New info | Scientific R. | Debiasing | 2-edits | 3-edits | 4-edits |
| **InComeS** | | 57.76 | 56.46 | 46.14 | 53.93 | 52.79 | 52.73 |
| - w/o zero-gist | | 56.85 | 57.46 | 45.72 | 47.48 | 41.71 | 32.57 |
| - w/ full model | Llama-3.2-1B | 56.25 | 54.28 | 38.31 | 51.91 | 52.59 | 52.53 |
| - w/o loss on query | | 58.89 | 57.49 | 46.25 | 49.63 | 46.44 | 38.28 |
| - w/ golden loss | | 56.46 | 53.62 | 43.05 | 55.39 | 50.24 | 40.06 |

Table 4: Ablation and Analysis experiments, the edit batch size is 100 for all results.

**Deciding inference temperature**  Applying a small temperature to the gist cross-attention sharpens the probability distribution over the gist KV caches, which facilitates the model's ability to retrieve the correct information. We determine the appropriate temperature based on entropy, which has been shown to be an important factor in attention mechanisms (Zhang et al., 2024b). Specifically, we aim to keep the cross-attention entropy close to its optimal value, which occurs when it only needs to attend to one edit. To achieve this, we select 1000 instances from ZsRE (Levy et al., 2017) and calculate the entropy of the edit batch size 1. We then calculate the entropy for larger edit batch sizes (10, 20, 40, 60, 80, 100, 200, 300) and find the temperatures that align their entropy with the optimal case via gradient descent [5]. We report the calculated temperature in Figure 4a. As expected, the temperature decreases as the edit batch size increases, but interestingly, it gradually converges to a specific value. Specifically, layers 9, 10, and 14 finally converge to around 0.5. To encourage more decisive selection, we slightly lower this value and set the temperature to $T = 0.45$.

**Information flow on tokens**  We further investigate the cross-attention patterns to understand how the model performs context selection. We measure the zero-gist and golden-gist probability (Figure 4c), and cross-attention entropy (Figure 4d) of each token from two representative examples containing a correctly ("success") and a wrongly ("fail") predicted instance using Llama-3.2-1B. As expected, the golden gist probability from the correctly predicted instance generally exceeds that of the failed one ("Golden prob - success" and "Golden prob - fail" line in Figure 4c). Notably, for all cases, the token at position zero allocates low probabilities to both golden and zero gists, while having high entropy, indicating that the model is "taking the average" of all gist representations at this beginning token. The dominance of zero-gist in later positions demonstrates that the model learns to "adaptively attend to edit information."

**Imposing golden loss on training**  As the golden gist representation is available for each training instance, it is natural to introduce an auxiliary loss to encourage correct selection in the cross-attention mechanism. We incorporate this additional loss in our experiments and report the result

---

[5]Note that all entropy is calculated in a way that the query and answer part are just a copy of the context

| Model | InComeS-Compression | ICL-Prefilling | FT-M | LoRA (Hu et al., 2022) | MEMIT (Meng et al., 2023) | EMMET (Gupta et al., 2024b) |
|---|---|---|---|---|---|---|
| Llama-3.2-1B | **0.0326** | 0.8934 | 3.4124 | 33.5412 | 112.2238 | 158.6524 |
| Qwen2.5-7B | **0.1071** | 0.9082 | 28.6132 | 122.1876 | 423.4823 | 512.4235 |

Table 5: Measured time (seconds) for 100 edits.

| Number of edits | Model | Encoding | | Decoding | |
|---|---|---|---|---|---|
| | | InComeS-Compression | ICL-Prefilling | InComeS-Selection | ICL-Generation (with prefilled cache) |
| 1k | Llama-3.2-1B | **0.2108** | 1.0413 | **0.0274** | 0.1555 |
| 2k | | **0.4051** | 1.2165 | **0.0297** | 0.3545 |

Table 6: Scaled efficiency comparison (seconds) between InComeS and ICL.

as "w/ golden loss" in Table 4. In the analysis in Figure 4, we use a suffix of "- t" to denote this setting. Incorporating the auxiliary loss leads the model to assign higher probabilities to the golden gist compared to training without this loss. Interestingly, it also increases the cross-attention entropy, probably because the model is explicitly encouraged to make selections during training. However, despite the increase in golden-gist probabilities, this approach does not yield clear performance improvements and even results in declines in some cases. This suggests that the model may develop its own context selection strategies, which do not always align with focusing all attention on the golden edit information.

**Side effect analysis** The side effect analysis is provided in Appendix E.2.

### 4.4 EFFICIENCY ANALYSIS

Finally, we present the efficiency analysis for our method. By default, the individual edit length is around 10 to 11. We first compare the efficiency of our method with the efficiency of other knowledge editing methods. Table 5 reports the time required to perform 100 edits for each method. Our method has significantly better efficiency than the other presented editing methods. Additionally, compared to ICL, our approach only needs to maintain the KV cache of the gist representations from the deeper half layers, resulting in substantially lower memory cost. To verify our method's superiority in efficiency on long context, we further conduct experiments on scaled context length (Table 6). The result demonstrates the efficiency advantage of our method in both the encoding and decoding stages. More detailed analysis can be found in Appendix E.3.

## 5 RELATED WORK

The area of knowledge editing (or model editing) has experienced a thriving development in recent years. Researchers have explored various directions in this area. One typical direction is to adopt external memory for the edits. The memory formats applied by different researchers are diverse. Methods like SERAC (Mitchell et al., 2022b), IKE (Zheng et al., 2023), MeLLo (Zhong et al., 2023a) adopt *explicit non-parametric memory*, which stores specific edit instances, and a retriever that is responsible for recalling relevant edits from the memory. For example, IKE uses KNN, and SERAC applies a trained classifier. Another line of work applies *implicit parametric memory*. CaliNET (Dong et al., 2022), T-Patcher (Huang et al., 2023) embeds the knowledge into a fixed number of neurons and adds them to the model. GRACE (Hartvigsen et al., 2022) adopts a discrete key-value codebook with the value optimized for the desired knowledge. MELO (Yu et al., 2024) applies dynamic LoRA blocks and indexes them via an internal vector database. KE (De Cao et al., 2021), MEND (Mitchell et al., 2022a) train a separate meta-model for editing. Another popular direction is to merge knowledge into the model directly. Methods like KN (Dai et al., 2022), ROME (Meng et al., 2022), R-ROME (Gupta et al., 2024a), MEMIT (Meng et al., 2023), PMET (Li et al., 2023), CoachHooK (Li et al., 2024), and AlphaEdit (Fang et al., 2024) perform editing by tweaking the located FFN part of the model directly. However, some studies reveal that these methods could potentially bring about side effects in the original model (Gu et al., 2024; Pinter & Elhadad, 2023), leaving the real effectiveness of these methods to be further investigated.

## 6 CONCLUSION

In this paper, we propose InComeS, a scalable model editing method that integrates compression and selection mechanisms directly into the LLMs. InComeS adopts a context compression technique to condense the editing context to KV representations on top of the introduced gist tokens and takes advantage of the compressed KVs to efficiently retrieve the relevant editing context information. Experiments on four different and complex editing settings demonstrate the superiority of our method for comprehensively editing. Further Analysis and ablations validate each component of InComeS and demonstrate the advantage of our method in efficiency.

## 7 ETHICS STATEMENT

The goal of our method is to conveniently and flexibly edit LLMs' behavior for reasonable and benign purposes, such as providing new relevant information or fixing false or inaccurate responses. However, we caution readers that our method should not be used for any malicious or offensive purpose, including but not limited to, political authoritative facts, rumors, discriminatory statements, etc. It is worth reminding that the safe and responsible application of our method is really important. None of the offensive, toxic, or malicious editing should be allowed.

## 8 REPRODUCIBILITY STATEMENT

The code repository of this project will be released, and the link will be included in the Introduction section of this paper.

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

## A    The Use of Large Language Models (LLMs)

The LLMs are only used for polishing the writing of this paper (word choosing, rephrasing, etc).

## B    Limitations

**Model scale & Architecture**    Due to the limited computational resources, we only extend the model size to 7B and leave the larger model size to future work. We are aware that the original gisting work (Mu et al., 2023) conducts experiments on three model architectures, i.e., encoder-decoder, encoder-only, and decoder-only. In this work, we determine to focus on the decoder-only autoregressive architecture as it is the structure used by most of the popular models nowadays (Yang et al., 2024; OpenAI, 2023; DeepSeek-AI et al., 2024).

**Compression rate**    In this work, we maintain the compression rate to roughly around 12:1 (one edit, which contains around 12 tokens for all testing datasets except DUNE (Akyürek et al., 2023), corresponds to one gist token), as one edit represents a fine-grained piece of information. However, we believe it is necessary to investigate the impact of lowering the compression rate, since it potentially helps extend the length of a single edit (Deng et al., 2024).

**Task variety**    InComeS can accept any input that follows natural language form. This flexibility gives it the potential to tackle many other tasks beyond knowledge editing. For example, long context language modeling, retrieval-augmented generation, etc. Due to the limited space of the main body of the paper, we first verify the effectiveness of our method on model editing and leave the investigation of other tasks to future work.

| Method | Model | WikiData$_{counterfact}$ | | | ZsRE-extended | | |
|---|---|---|---|---|---|---|---|
| | | Edit Success | Portability | Locality | Edit Success | Portability | Locality |
| Base | | 21.28 / 21.28 | 19.73 / 19.73 | - | 30.06 / 30.06 | 40.17 / 40.17 | - |
| FT-M | | 97.02 / 94.58 | 53.43 / 47.51 | 49.01 / 20.54 | 99.81 / 95.94 | 62.80 / 54.84 | 75.59 / 59.28 |
| LoRA (Hu et al., 2022) | | 98.91 / 82.61 | 52.87 / 43.84 | 23.31 / 14.94 | 99.86 / 93.18 | 57.43 / 44.85 | 34.94 / 37.41 |
| ROME (Meng et al., 2022) | | 94.33 / - | 40.44 / - | 26.13 / - | 95.41 / - | 46.04 / - | 40.19 / - |
| R-ROME (Gupta et al., 2024a) | | 94.31 / - | 40.65 / - | 25.83 / - | 95.29 / - | 46.46 / - | 39.95 / - |
| MEMIT (Meng et al., 2023) | | - / 66.94 | - / 23.51 | - / 11.12 | - / 58.79 | - / 25.68 | - / 30.10 |
| EMMET (Gupta et al., 2024b) | | - / 27.02 | - / 08.86 | - / 100.0 | - / 15.96 | - / 06.84 | - / 100.0 |
| MEND (Mitchell et al., 2022a) | Llama-3.2-1B | - / 26.66 | - / 21.06 | - / 13.54 | - / 43.33 | - / 30.77 | - / 38.75 |
| GRACE (Hartvigsen et al., 2022) | | 33.27 / 25.06 | 14.33 / 10.51 | 28.71 / 11.43 | 32.00 / 24.44 | 12.73 / 10.91 | 24.12 / 10.12 |
| KN (Dai et al., 2022) | | 20.60 / - | 17.16 / - | 19.46 / - | 16.01 / - | 06.70 / - | 21.23 / - |
| IKE (Zheng et al., 2023) | | 61.70 / - | 45.55 / - | 48.80 / - | 59.15 / - | 57.39 / - | 40.21 / - |
| SERAC (Mitchell et al., 2022b) | | 89.56 / 78.32 | 60.56 / 46.45 | 45.67 / 39.36 | 92.69 / 89.61 | 66.59 / 63.60 | 48.96 / 41.32 |
| ICL | | 93.31 / 82.95 | 65.81 / 49.75 | 52.70 / 45.59 | 68.86 / 60.84 | 62.19 / 55.58 | 62.78 / 59.43 |
| **InComeS** | | 91.16 / 76.81 | 65.15 / 45.66 | 55.22 / 53.97 | 97.22 / 87.09 | 70.70 / 52.23 | 61.33 / 59.66 |
| Base | | 22.35 / 22.35 | 21.46 / 21.46 | - | 36.21 / 36.21 | 43.86 / 43.86 | - |
| FT-M | | 98.93 / 90.18 | 49.39 / 43.13 | 15.73 / 12.93 | 99.51 / 92.60 | 50.04 / 46.41 | 18.00 / 27.27 |
| LoRA (Hu et al., 2022) | | 77.31 / 72.22 | 37.04 / 31.65 | 0.380 / 02.24 | 86.88 / 77.78 | 28.61 / 24.13 | 1.290 / 0.330 |
| ROME (Meng et al., 2022) | | 92.69 / - | 40.25 / - | 38.76 / - | 97.86 / - | 50.43 / - | 51.38 / - |
| R-ROME (Gupta et al., 2024a) | | 92.59 / - | 40.15 / - | 38.70 / - | 97.89 / - | 50.47 / - | 51.31 / - |
| MEMIT (Meng et al., 2023) | | - / 91.16 | - / 39.85 | - / 25.85 | - / 93.28 | - / 49.97 | - / 51.85 |
| EMMET (Gupta et al., 2024b) | Qwen2.5-7B | - / 88.05 | - / 39.37 | - / 100.0 | - / 93.64 | - / 48.44 | - / 100.0 |
| MEND (Mitchell et al., 2022a) | | - / 35.13 | - / 15.29 | - / 07.29 | - / 50.91 | - / 38.83 | - / 47.12 |
| GRACE (Hartvigsen et al., 2022) | | 31.34 / 33.77 | 25.60 / 18.55 | 19.19 / 07.29 | 33.27 / 26.79 | 14.35 / 11.25 | 12.31 / 23.45 |
| KN (Dai et al., 2022) | | 36.33 / - | 29.60 / - | 32.79 / - | 14.49 / - | 8.49 / - | 33.21 / - |
| IKE (Zheng et al., 2023) | | 96.40 / - | 75.33 / - | 46.98 / - | 99.75 / - | 83.17 / - | 43.19 / - |
| SERAC (Mitchell et al., 2022b) | | 91.79 / 80.68 | 51.12 / 41.26 | 37.84 / 35.10 | 91.12 / 82.56 | 62.41 / 52.63 | 40.56 / 42.46 |
| ICL | | 90.24 / 85.28 | 66.99 / 51.66 | 52.63 / 40.97 | 71.75 / 71.57 | 66.10 / 64.57 | 51.43 / 47.12 |
| **InComeS** | | 90.96 / 71.44 | 66.69 / 47.93 | 52.04 / 42.02 | 97.95 / 91.29 | 75.63 / 61.22 | 57.16 / 59.21 |

Table 7: More results for $WikiData_{counterfact}$ (Cohen et al., 2024; Zhang et al., 2024a) and ZsRE-extended (Yao et al., 2023; Zhang et al., 2024a). The data format of each cell is in "single-edit result / 100-edits result".

## C  TRAINING DETAILS

InComeS is trained on around 1.5 billion tokens, which mainly come from summarization and QA datasets. Specifically, for summerization datasets, we select $4.5e^6$ instances from S2ORC (Lo et al., 2020), $1.15e^6$ instances from AG News Corpus [6]; and for QA datasets, we use squad (Rajpurkar et al., 2016), a modified version [7] of the natural question dataset (Kwiatkowski et al., 2019), Open-BookQA (Mihaylov et al., 2018), QASC (Khot et al., 2020), MedMCQA (Pal et al., 2022), and NetEval (Miao et al., 2023). We also include the training split of ZsRE (Levy et al., 2017), COUNTERFACT (Meng et al., 2023), $Wiki_{counterfact}$ from EasyEdit framework (Wang et al., 2023).

We use a cosine linear-warmup scheduler for both models, with a maximum learning rate $1e^{-5}$ and a minimum learning rate $1e^{-6}$ for Llama-3.2-1B and a maximum $5e^{-6}$ and a minimum $1e^{-6}$ for Qwen2.5-7B. To improve the model's robustness and sample them at a predefined rate during training, the batch size is dynamically sampled from a predefined set rather than a fixed number. Specifically, the predefined set for batch size is 8, 16, 32, 64, and 128, and their corresponding sample rates are 0.05, 0.05, 0.05, 0.15, and 0.7. We adopt DeepSpeed (Rajbhandari et al., 2020; 2021; Ren et al., 2021) and Liger Kernel (Hsu et al., 2024) with 8 Nvidia GPUs for distributed training. Overall, the training takes around 11 hours for Llama-3.2-1B and 35 hours for Qwen2.5-7B.

## D  EXPERIMENT DETAILS

### D.1  DATASETS

**MQuAKE**  The dataset MQuAKE (Zhong et al., 2023b) (Multi-hop Question Answering for Knowledge Editing) is constructed based on Wikidata and contains question answering instances that require 2-hop, 3-hop, and 4-hop reasoning. In the experiment, we use the latest version of

---

[6]https://huggingface.co/datasets/sentence-transformers/agnews
[7]https://huggingface.co/datasets/LLukas22/nq-simplified

the dataset [8], which fixes the knowledge conflict problem for the old version multi-edit subset, and report the accuracy for each query.

**DUNE**    DUNE (Akyürek et al., 2023) is a benchmark designed for edits in natural language form. It evaluates the model's capability of conducting natural language edits through four aspects: scientific reasoning, arithmetic reasoning, new information, and debiasing. As illustrated in Table 2 of (Akyürek et al., 2023), the arithmetic reasoning edits do not follow natural language form as other subsets do and cannot represent a complete piece of instruction, therefore, we do not include it in our experiment.

**WikiData$_{counterfact}$**    The WikiData$_{counterfact}$ (Cohen et al., 2024; Zhang et al., 2024a) collects triplets from top-viewed pages from Wikipedia and contains portability (ripple-effect (Cohen et al., 2024)) instances to test whether the output to the input relevant to the edits is changed as well. Specifically, the portability evaluates the post-edited model from three aspects, including logical generalization, subject aliasing, and reasoning.

**ZsRE-extended**    The extended version of ZsRE (Zhang et al., 2024a; Yao et al., 2023) is constructed based on the original ZsRE (Levy et al., 2017), which is a dataset that focuses on the QA task. The extended version introduces a portability test (Yao et al., 2023), including inverse relation, one-hop reasoning, and subject aliasing.

**COUNTERFACT**    COUNTERFACT (Meng et al., 2023) is a dataset that concentrates on counterfactual information, which typically receives a lower prediction score than accurate facts. It constructs out-of-scope data by substituting the subject entity with a comparable description that has the same predicate.

### D.2    Evaluation metrics

This section explains the evaluation metrics used in the extended ZsRE(Yao et al., 2023; Zhang et al., 2024a) and Wiki$_{counterfact}$ (Cohen et al., 2024; Zhang et al., 2024a). Generally, they adopt four metrics: reliability, generality, portability, and locality. Given an initial base model $f_\theta$, a post-edit model $f_{\theta'}$, and a set of edit instances $(x_t, y_t) \in \{(x_t, y_t)\}$, the reliability is computed as the average accuracy of the edit cases:

$$\mathbb{E}_{(x_t, y_t) \in \{(x_t, y_t)\}} \{\arg\max_y f_{\theta'}(y|x_t) = y_t\} . \tag{9}$$

The editing should also edit the equivalent neighbor of the instance $(x'_t, y'_t) \in N(x_t, y_t)$ (e.g. rephrased descriptions). This metric is named generality and is evaluated by the average accuracy on the neighbors of the edit cases:

$$\mathbb{E}_{(x'_t, y'_t) \in \{N(x_t, y_t)\}} \{\arg\max_y f_{\theta'}(y|x'_t) = y'_t\} . \tag{10}$$

Beyond simple rephrasing, the editing is also supposed to affect other sophisticatedly related instances $(x''_t, y''_t) \in P(x_t, y_t)$. For example, instances that require reasoning, logical generalization over the edits. This metric is defined as portability:

$$\mathbb{E}_{(x''_t, y''_t) \in \{P(x_t, y_t)\}} \{\arg\max_y f_{\theta'}(y|x''_t) = y''_t\} . \tag{11}$$

Despite the editing, those instances that are irrelevant to the edit cases $(\hat{x}_t, \hat{y}_t) \in \{O(x_t, y_t), f_\theta(x_t) = y_t\}$ should not be affected. This evaluation is called locality (also known as specificity) and is measured by the proportion of unchanged predictions between the initial model and the post-edit model:

$$\mathbb{E}_{(\hat{x}_t, \hat{y}_t) \in \{O(x_t, y_t)\}} \{f_{\theta'}(\hat{x}_t) = f_\theta(\hat{x}_t)\} . \tag{12}$$

For the extended ZsRE (Yao et al., 2023; Zhang et al., 2024a) and Wiki$_{counterfact}$ (Cohen et al., 2024; Zhang et al., 2024a), we follow the setting in the original paper and combine reliability and generality to the Edit Success rate.

---

[8]"MQuAKE-CF-3k-v2.json" in https://github.com/princeton-nlp/MQuAKE

### D.3 BASELINE IMPLEMENTATION DETAILS

Unless otherwise specified, the baselines are implemented by using the EasyEdit framework (Wang et al., 2023).

**Fine-tuning** We follow the procedure implemented in previous work (Meng et al., 2022; 2023; Yao et al., 2023; Zhang et al., 2024a) to fine-tune a specific layer from the model. We select layer 13 for Llama-3.2-1B and layer 27 for Qwen2.5-7B. For both models, we adopt the learning rate of $5e^{-4}$ and the number of optimization steps 25.

**LoRA** For both models, we use LoRA (Hu et al., 2022) to update the query and key projection matrix of the models, with rank set to 8, $\alpha$ set to 32, the dropout rate 0.1, and the learning rate $5e^{-3}$. The number of updating steps is set to 70 for Llama-3.2-1B and 60 for Qwen2.5-7B.

**ROME** ROME (Meng et al., 2022) treats the FFN part of the LLMs as a key-value association and updates a pre-located layer by directly inserting an optimized key-value pair. We update the layer 5 for both Llama-3.2-1B and Qwen2.5-7B, and adopt 25 optimization steps for Llama-3.2-1B and 20 optimization steps for Qwen2.5-7B, with both learning rate $5e^{-5}$.

**R-ROME** R-ROME (Gupta et al., 2024a) is another version of ROME (Meng et al., 2022) with modified code implementation. We use the same hyperparameters as ROME.

**KN** KN (Dai et al., 2022) hypothesize that factual knowledge is stored in FFN memories and expressed by knowledge neurons. For both models, we use the threshold of 0.2 for knowledge attribution scores and 0.4 for the threshold of the prompts sharing percentage.

**GRACE** GRACE (Hartvigsen et al., 2022) adopts a discrete codebook to memorize the edits as key-value pairs. We set the location of the codebook layer 13 and 18 for Llama-3.2-1B and Qwen2.5-7B, respectively. Surprisingly, the $\epsilon$ value used in the original paper (1-3) seems insufficient for the complex editing experiments in this paper. Therefore, we increase it to 50. The number of optimization steps for the value vector is set to 100.

**IKE** IKE (Zheng et al., 2023) maintains an explicit memory for edits and retrieves them via K-nearest neighbors. The retrieved edits are then used to construct demonstrations, which are then prefixed to the input to edit the behavior. In the experiments, we set $K = 16$.

**MEND** MEND (Mitchell et al., 2022a) trains an additional meta-network to predict a new rank-one update to the input gradient. In this paper, we train each model using ZsRE (Levy et al., 2017) and COUNTERFACT (Meng et al., 2023), and adopt the ZsRE-trained model to MQuAKE and the extended ZsRE and COUNTERFACT-trained model for Wiki$_{counterfact}$.

**SERAC** SERAC (Mitchell et al., 2022b) employs an explicit edit-instance memory, an additional trained scope classifier, and a trained counterfactual model. The scope classifier is responsible for determining whether an input is relevant to the edits in the memory. The input is fed to the counterfactual model once the input is deemed as relevant to memorized edits and the original model otherwise. We use the distilbert-base-cased (Sanh et al., 2019) model as the scope classifier, and train it using the training set of ZsRE (Levy et al., 2017) and COUNTERFACT (Meng et al., 2023) from EasyEdit (Wang et al., 2023). Following (Akyürek et al., 2023), we use instruction-tuned models (Llama-3.2-1B-Instruct[9] and Qwen2.5-0.5B[10]) for the counterfactual model.

**MEMIT** MEMIT (Meng et al., 2023) is the extension of ROME (Meng et al., 2022) that supports a batch of edits at a time. Unlike ROME, which only updates a single pre-located layer, MEMIT spreads the update to a set of identified layers. We apply changes to layers 4-8 for both models. Following the settings in the original paper, we set $\lambda$, the hyperparameter that balances the weighting of new and old associations, to $1.5 \times 10^4$.

---

[9]https://huggingface.co/meta-llama/Llama-3.2-1B-Instruct

[10]https://huggingface.co/Qwen/Qwen2.5-0.5B-Instruct

| Method | Model | Single Editing | | | Batch Editing | | |
|--------|-------|---------|---------|---------|---------|---------|---------|
| | | 2-edits | 3-edits | 4-edits | 2-edits | 3-edits | 4-edits |
| Base | | 41.79 | 43.51 | 31.58 | 41.79 | 43.51 | 31.58 |
| RAG | Llama-3.2-1B | 59.23 | 59.00 | 51.63 | 40.65 | 46.78 | 43.58 |
| ICL | | 59.23 | 59.00 | 51.63 | 50.06 | 49.86 | 42.37 |
| **InComeS** | | **71.19** | **72.17** | **72.62** | **53.93** | **52.79** | **52.73** |
| Base | | 44.08 | 44.14 | 30.62 | 44.08 | 44.14 | 30.62 |
| RAG | Qwen2.5-7B | 69.76 | 76.91 | 74.54 | 42.91 | 46.37 | 46.15 |
| ICL | | **69.76** | **76.91** | 74.54 | 53.53 | 50.54 | 44.77 |
| **InComeS** | | 66.46 | 71.24 | **76.54** | **55.13** | **53.48** | **47.91** |

Table 8: Results for RAG on MQuAKE (Zhong et al., 2023b).

**EMMET** EMMET (Gupta et al., 2024b) is an unification of ROME (Meng et al., 2022) and MEMIT (Meng et al., 2023). Similar to ROME, we edit the layer 5 for both models, and set $\lambda = 1e^5$.

**RAG** We adopt bge-base-en-v1.5 (Xiao et al., 2023) as our retriever. For batch editing, we treat the corresponding batch number of edits as our corpus, and retrieve the 10 most relevant edits for each testing query.

**InComeS** We apply cross-attention operations only for the second half of the model's layers since we found that the gist KV cache from the first half is not informative enough to allow effective edit selection. For the calculation of cross-attention during inference, we adopt a temperature of $T = 0.45$ to the logits before softmax, which is found to be helpful for effective editing.

# E  FURTHER ANALYSIS

## E.1  RAG VS. INCOMES

**Problem settings** InComeS and RAG target different problems. In RAG, the query is given and used to retrieve the relevant documents via a retrieval model before decoding. However, InComeS does not make such an assumption, and it "edits" the model with the provided knowledge before seeing any actual queries.

**Methodology** From the perspective of methodology, InComeS conducts selection mechanisms in a way that is better integrated into the LM and with a finer granularity compared to RAG methods. First, RAG inputs the full query for retrieval to select relevant documents, and this process needs an extra retrieval model; in contrast, our method requires no extra models or retrieval steps and directly integrates the selection mechanism into our base LM. Moreover, our method dynamically performs selection for each token individually, which has a much finer granularity than the query-based selection in RAG. This supports a wider range of applications.

**Experiments on Multi-hop edits** Despite the differences between our method and RAG, we compare our method with RAG to demonstrate our method's superiority over the complex editing scenarios (Table 8). InComeS outperforms RAG in almost all cases for both single editing and batch editing. Note that the result of RAG is the same as the result of ICL in single editing, which does not need retrieval.

## E.2  SIDE EFFECT

We present a side effect analysis of our method in this section (Table 9). We test the editing side effect under three different numbers of edits (0, 0.1k, 1k) on MMLU (Hendrycks et al., 2021) benchmark, which consists of 57 tasks across 4 domains, namely Social science, Humanities, STEM, and others. The results indicate that increasing the number of edits does not significantly harm the model's general capability (lines 3 - 5 and 7 - 9 in Table 9), demonstrating the potential scalability of our method. The continuous pre-training brings an inevitable modest side effect to the model (lines 2 - 3 and 6- 7 in Table 9).

| Method | Model | Social Sciences | Humanities | STEM | Other | Average |
|--------|-------|-----------------|------------|------|-------|---------|
| Base | | 25.09 | 27.31 | 25.87 | 27.49 | 26.44 |
| InComeS - w/ no edits | Llama-3.2-1B | 23.18 | 25.91 | 22.66 | 26.06 | 24.45 |
| InComeS - w/ 0.1k edits | | 22.45 | 25.01 | 22.66 | 24.62 | 23.68 |
| InComeS - w/ 1k edits | | 22.91 | 25.30 | 22.73 | 24.11 | 23.76 |
| Base | | 72.72 | 69.19 | 63.42 | 69.72 | 68.76 |
| InComeS - w/ no edits | Qwen2.5-7B | 68.06 | 65.68 | 59.58 | 66.39 | 64.93 |
| InComeS - w/ 0.1k edits | | 64.64 | 60.65 | 56.84 | 63.39 | 61.38 |
| InComeS - w/ 1k edits | | 63.86 | 62.08 | 57.58 | 62.88 | 61.60 |

Table 9: Side effect evaluation on MMLU (Hendrycks et al., 2021).

### E.3 EFFICIENCY ANALYSIS

Compared to many traditional editing methods that require model backward calculation, our method only requires one single forward pass for each editing context. In comparison to ICL, which needs to encode the entire concatenated edit context, our approach enables parallel encoding of multiple edits, leading to great efficiency gains for the encoding (prefilling) stage. In addition, the compressed context also accelerates the decoding phase compared to the ICL decoding with prefilled KV cache. Here, we provide an analysis of our method's efficiency advantage over traditional ICL for both the encoding and decoding stages.

**Encoding** Assume we have $N$ edits and each edit has a Length of $L$. For ICL prefilling, it has to encode the whole sequence with length $N \times L$. However, for InComeS, each edit is processed individually, and it encodes edits in parallel. In this case, it encodes a batch of $N$ edits with length $L$. Thanks for the highly optimized GPU parallel computation, such a feature approximately reduces the time consumption by $N$ times.

**Decoding** Suppose we have $N$ compressed gists, which corresponds to $N$ individual edits with length $L$ and $N \times L$ tokens whose KV caches have been prefilled. For each decoding position, the ICL self-attention needs to compute a matrix with size $1 \times N \times L$. However, InComeS only needs to calculate the gist cross-attention matrix with size $1 \times N$. This roughly accelerates the decoding by $L$ times.

