# OpenReview forum: "InComeS: Integrating Compression and Selection Mechanisms into LLMs for Efficient Model Editing"
_ICLR.cc/2026/Conference — ICLR 2026 Conference Withdrawn Submission_

### Official Review · Reviewer_2yPy · 2025-10-21

**Soundness:** 3
**Presentation:** 3
**Contribution:** 2
**Rating:** 4
**Confidence:** 4

**Summary:**

This paper introduces InComeS, a novel framework for efficient model editing in Large Language Models (LLMs). It addresses a key limitation of In-Context Learning (ICL)-based editing methods, whose performance and efficiency degrade with a large number of edits due to the finite context window. The proposed solution involves compressing each edit context into a special "gist token's" KV cache, bypassing the context length constraint. Furthermore, the authors incorporate cross-attention modules to enable the model to dynamically select the most relevant information from a pool of these compressed gist tokens. Experiments across various complex editing benchmarks (e.g., multi-hop, natural language edits) demonstrate that InComeS outperforms existing methods in effectiveness and efficiency.

**Strengths:**

The paper clearly identifies a significant and practical challenge in model editing and propose InComeS a flexible framework that enhances LLMs’ ability to process editing contexts through explicit compression and selection mechanism.

The Gist token used in Editing is very interesting. And editing the attention module is also novel.

**Weaknesses:**

* How can the GIST token be effectively trained? Furthermore, once trained, what metrics should be used to evaluate the generalization capability of the GIST token?
﻿
* The training process requires approximately 11 hours for Llama-3.2-1B and 35 hours for Qwen2.5-7B. Considering the performance gains achieved, how does the efficiency of this approach compare to other model editing techniques, such as In-Context Learning (ICL)?
﻿
* As your experiments indicate, simple fine-tuning (FT) can yield strong results. However, incorporating a more detailed analysis of locality could further strengthen the evaluation, given that locality preservation is a key consideration in model editing methods.

**Questions:**

* Under what circumstances is the GIST token applied? Is it used for every inference instance, or is it selectively activated based on specific input criteria?
 * Given that model editing typically prioritizes efficiency, what is the rationale behind adopting a teacher-student training framework, especially considering its apparently substantial computational cost?

---

> ### Author Response · Authors · 2025-11-21
>
> Thank you very much for your thorough review, and we would like to respond to your concerns as follows:
>
> > - How can the GIST token be effectively trained? Furthermore, once trained, what metrics should be used to evaluate the generalization capability of the GIST token?
>
> The meta-training process is discussed in section 3.3 and figure 2. The heuristic is that we want the model with the compressed gist representations to behave similarly to the case when full non-compressed original texts can be seen. Following this motivation, we apply a frozen teacher model and use its context-aware output to distill our gist model.
>
> We believe the generalization capability can be evaluated via investigating the selection pattern of the cross attention, which can be represented by the **golden gist probability** and the **zero-gist probability** ("Information loss on tokens" in section 4.3, Figure 4c). The results reveal that the golden gist probability from the correctly predicted instance generally exceeds that of the failed one (”Golden prob - success” and ”Golden prob - fail” line in Figure 4c), which demonstrates the effectiveness of the selection mechanism. In addition, the dominance of zero-gist probability indicates that the model learns to ”adaptively attend to edit information”, which we believe is evidence for the generalization capability.
>
>
> > - The training process requires approximately 11 hours for Llama-3.2-1B and 35 hours for Qwen2.5-7B. Considering the performance gains achieved, how does the efficiency of this approach compare to other model editing techniques, such as In-Context Learning (ICL)?
>
> Generally, our approach has three stages: meta-training (one-time training), compressing or editing (compressing the context information to gist representations, corresponding to the prefilling in ICL), and inference selection (cross attention between query tokens and gist representations, corresponding to the decoding in ICL).
>
> The training process used in our method can be understood as a **meta-training** process, meaning that once trained, the model can be applied to different data distributions just like ICL. Unlike the traditional training process that is conducted once for every data distribution, the meta-training adopted in our method is a "one take for all" approach.
>
> The efficiency analysis of our method and ICL is given in Section 4.4, Table 5, and Table 6. The results demonstrate a great inference efficiency gain of our method over both the editing (prefilling) and the selection phase. Given the inference efficiency gain and corresponding performance gain over complex editing scenarios, we believe the one-time meta-training cost is acceptable.
>
>
> > - As your experiments indicate, simple fine-tuning (FT) can yield strong results. However, incorporating a more detailed analysis of locality could further strengthen the evaluation, given that locality preservation is a key consideration in model editing methods.
>
> We fully agree with your point and apologize for the caused misunderstanding. The locality is included in Table 7, which is the complete version of Table 3 (stated in line 323). Table 3 is simplified due to limited space. The results in Table 7 show the superiority of our method in the locality. We have added another reminder to the caption of Table 3.
>
>
> > - Under what circumstances is the GIST token applied? Is it used for every inference instance, or is it selectively activated based on specific input criteria?
>
> Theoretically, the application of gist tokens is actively determined by users. But in this work, we follow the simple heuristics and assign one gist for one edit. Our method is as flexible as ICL and does not have any specific requirement for the input format. We believe this is a big advantage over many other editing methods that are based on triplet input and do not support natural language edits ([1], [2], [3], etc).
>
>
> > - Given that model editing typically prioritizes efficiency, what is the rationale behind adopting a teacher-student training framework, especially considering its apparently substantial computational cost?
>
> The distillation framework is to let the model with the compressed gist representations behave similarly to the case when full non-compressed original texts can be seen (the ICL setting).
>
> As mentioned in the response to your second concern, the meta-training process happens only once. The model is free to apply to any editing data after the one-time meta-training. For example, the results of our method for different editing scenarios in Tables 1, 2, and 3 are from the same model trained from a single meta-training. Given the demonstrated efficiency gain (Tables 5 and 6) and performance gain (Tables 1, 2, etc), we believe the one-time meta-training cost is acceptable.
>
>
>
> [1] MASS-EDITING MEMORY IN A TRANSFORMER
>
> [2] Locating and Editing Factual Associations in GPT
>
> [3] Lifelong Knowledge Editing for LLMs with Retrieval-Augmented Continuous Prompt Learning

---

> > ### Comment · Reviewer_2yPy · 2025-11-25
> >
> > Thank you for your reply. Considering the training cost, effectiveness, and mechanism of GIST tokens, I will keep the score unchanged.
> >
> > I recommend simplifying the training of GIST tokens and applying them in a more adaptive manner. Don't just Distillation same as Learning to Compress Prompts with Gist Tokens. Or explain why Distillation is necessary.

---

> > > ### Author Response · Authors · 2025-11-26
> > >
> > > Thank you for your comments.
> > >
> > > We will run ablation experiments to verify the necessity of the distillation and post it here for your reference.

---

> > > ### Author Response · Authors · 2025-11-30
> > >
> > > Thank you for your feedback. To verify the necessity of the distillation, we run ablation experiments for the two components related to the distillation, which are the KL divergence loss and the distillation weights for the cross-entropy loss. The results are shown below ("s" means "single editing", "b" means "batch editing"):
> > >
> > > MquAKE (Llama-3.2-1B):
> > > | Method          | 2-edits-s| 3-edits-s| 4-edits-s| 2-edits-b| 3-edits-b| 4-edits-b|
> > > | -------         | -------- | -------- | -------- | -------- | -------- | -------- |
> > > | InComeS         |**71.19** |**72.17** |**72.62** |**53.93** |**52.79** |**52.73** |
> > > | w/o kl loss     | 57.75    | 57.38    | 50.18    | 49.64    | 49.26    | 46.51    |
> > > | w/o ce weights  | 57.45    | 57.84    | 49.76    | 49.86    | 49.23    | 47.47    |
> > >
> > > The performance decreases when we eliminate either one of them. We believe this can prove the necessity of the distillation.

---

### Official Review · Reviewer_cAvh · 2025-10-30

**Soundness:** 2
**Presentation:** 2
**Contribution:** 2
**Rating:** 4
**Confidence:** 4

**Summary:**

The paper proposes InComeS, an in-context–style editing framework that (i) compresses each edit into a single “gist” token’s KV cache and (ii) equips the base LM with cross-attention modules that, at generation time, select among a pool of cached gists plus a special “zero-gist” option. Training uses token-wise reweighting—based on loss differences with/without edit context from a teacher model—together with a KL term to distill the teacher into the student. Experiments on MQuAKE, DUNE, WikiDataCounterfact and ZsRE-extended show competitive accuracy and efficiency versus ICL and a range of editing baselines.

**Strengths:**

(1) Compressing edits into re-usable gist KV caches and adding token-level cross-attention to select among them is clean; Zero-gist, serving as a “no-selection” option, reduces interference from the edit context on irrelevant tokens (see ablations) and complements the locality metric.
(2) Evaluates the effectiveness of the method across multiple scenarios, including multi-hop edits (MQuAKE), natural-language edits (DUNE), and ripple/portability settings (WikiDataCounterfact, ZsRE-extended).

**Weaknesses:**

(1) The paper does not include comparisons with recent strong editors such as memory based RECIPE[1] and ICL retriever based DR-IKE[2]. Without these, the empirical claims lack persuasiveness regarding true advances over contemporary methods.
[1]Lifelong Knowledge Editing for LLMs with Retrieval-Augmented Continuous Prompt Learning
[2]Dynamic Retriever for In-Context Knowledge Editing via Policy Optimization
(2) Despite criticizing ICL’s limitations, results show InComeS often performs on par with or worse than ICL—e.g., On multi-hop, InComeS underperforms ICL on Qwen2.5-7B for single 2- (66.46% vs 69.76%) and 3-hop (71.24% vs 76.91) settings (Table 1); on portability, InComeS is close to or slightly below ICL for Llama-3.2-1B and Qwen2.5-7B on batch editing results, such as WikiDatacounterfact Edit Success (71.44% vs 85.28%) and ZsRE-extended Portability (61.22% vs 64.57%) (Table 3).

**Questions:**

See Weaknesses

---

> ### Author Response · Authors · 2025-11-21
>
> Thank you very much for your review, and we would like to respond to your concerns as follows:
>
> > (1) The paper does not include comparisons with recent strong editors such as memory based RECIPE[1] and ICL retriever based DR-IKE[2]. Without these, the empirical claims lack persuasiveness regarding true advances over contemporary methods. [1]Lifelong Knowledge Editing for LLMs with Retrieval-Augmented Continuous Prompt Learning [2]Dynamic Retriever for In-Context Knowledge Editing via Policy Optimization
>
> Thanks for pointing this out. Please see the results as follows ("s" means "single editing", "b" means "batch editing"):
>
> MquAKE (Llama-3.2-1B):
> | Method   | 2-edits-s| 3-edits-s| 4-edits-s| 2-edits-b| 3-edits-b| 4-edits-b|
> | -------  | -------- | -------- | -------- | -------- | -------- | -------- |
> | base     | 41.79    | 43.51    | 31.58    | 41.79    | 43.51    | 31.58    |
> | RECIPE   | 58.44    | 54.69    | 53.33    | 50.71    | 47.34    | 39.39    |
> | DR-IKE   | 52.95    | 48.26    | 47.12    | 42.45    | 41.33    | 32.22    |
> | InComeS  |**71.19** |**72.17** |**72.62** |**53.93** |**52.79** |**52.73** |
>
> MquAKE (Qwen2.5-7B):
> | Method   | 2-edits-s| 3-edits-s| 4-edits-s| 2-edits-b| 3-edits-b| 4-edits-b|
> | -------  | -------- | -------- | -------- | -------- | -------- | -------- |
> | base     | 44.08    | 44.14    | 30.62    | 44.08    | 44.14    | 30.62    |
> | RECIPE   | 57.54    | 59.69    | 58.43    | 49.24    | 49.78    | 40.71    |
> | DR-IKE   | 60.95    | 54.66    | 51.22    | 44.35    | 39.99    | 39.77    |
> | InComeS  |**66.46** |**71.24** |**76.54** |**55.13** |**53.48** |**47.91** |
>
>
> DUNE (Llama-3.2-1B):
> | Method   | new info-s| scientific-s| de-biasing-s| new info-b| scientific-b| de-biasing-b|
> | -------  | --------  | --------    | --------    | --------  | --------    | --------    |
> | base     | 56.85     | 55.87       | 32.73       | 56.85     | 55.87       | 32.73       |
> | DR-IKE   | 58.55     | 56.22       | 40.99       | 54.76     | 55.23       | 35.67       |
> | InComeS  |**60.00**  |**58.17**    |**54.61**    |**57.76**  |**56.46**    |**46.14**    |
>
> DUNE (Qwen2.5-7B):
> | Method   | new info-s| scientific-s| de-biasing-s| new info-b| scientific-b| de-biasing-b|
> | -------  | --------  | --------    | --------    | --------  | --------    | -------- |
> | base     | 63.44     | 66.03       | 36.95       | 63.44     | 66.03       | 36.95    |
> | DR-IKE   | 64.88     | 66.26       | 40.12       | 63.55     | 65.33       | 38.22    |
> | InComeS  |**66.83**  |**68.02**    |**62.59**    |**65.61**  |**67.82**    |**56.69** |
>
> The results further demonstrate the effectiveness of the method. Note that there is no result for RECIPE[1] in the DUNE dataset, because it updates the model parameters based on the triplet knowledge statement, which makes it inapplicable to natural language edits. This is also an important advantage of our method.

---

> ### Author Response · Authors · 2025-11-21
>
> > (2) Despite criticizing ICL’s limitations, results show InComeS often performs on par with or worse than ICL—e.g., On multi-hop, InComeS underperforms ICL on Qwen2.5-7B for single 2- (66.46% vs 69.76%) and 3-hop (71.24% vs 76.91) settings (Table 1); on portability, InComeS is close to or slightly below ICL for Llama-3.2-1B and Qwen2.5-7B on batch editing results, such as WikiDatacounterfact Edit Success (71.44% vs 85.28%) and ZsRE-extended Portability (61.22% vs 64.57%) (Table 3).
>
> Thanks for your comments. Given this performance variance, we conduct additional experiments on a more advanced model (Qwen3-8B) to investigate the effectiveness of our method. Results are shown below ("s" means "single editing", "b" means "batch editing"):
>
> MquAKE (Qwen3-8B):
> | Method   | 2-edits-s| 3-edits-s| 4-edits-s| 2-edits-b| 3-edits-b| 4-edits-b|
> | -------  | -------- | -------- | -------- | -------- | -------- | -------- |
> | base     | 43.49    | 41.01    | 26.32    | 43.49    | 41.01    | 26.32    |
> | FT-M     | 51.32    | 50.39    | 45.22    | 48.11    | 44.24    | 41.39    |
> | LoRA     | 38.95    | 36.27    | 33.02    | 24.41    | 21.24    | 22.12    |
> | MEMIT    | 49.14    | 46.05    | 41.15    | 42.34    | 40.10    | 33.45    |
> | ICL      | 55.82    | 54.13    | 47.06    | 50.78    | 48.84    | 45.28    |
> | InComeS  |**56.36** |**58.71** |**59.93** |**53.42** |**50.90** |**50.36** |
>
> DUNE (Qwen3-8B):
> | Method   | new info-s| scientific-s| de-biasing-s| new info-b| scientific-b| de-biasing-b|
> | -------  | --------  | --------    | --------    | --------  | --------    | --------    |
> | base     | 82.99     | 86.62       | 31.07       | 82.99     | 86.62       | 31.07       |
> | FT-M     | 78.23     | 75.44       | 56.38       | 76.33     | 72.34       | 54.28       |
> | LoRA     | 71.23     | 69.99       | 43.23       | 73.13     | 64.59       | 42.22       |
> | SERAC    | 81.29     | 80.45       | 49.65       | 80.99     | 78.53       | 45.66       |
> | ICL      | 83.06     | 81.12       | 45.45       | 83.35     | 82.76       | 30.34       |
> | InComeS  | **83.17** | **85.36**   | **63.96**   | **84.17** | **87.57**   | **65.96**   |
>
> The results further prove the effectiveness of our method. Please note that the advantage of our method is not merely the performance, it also includes the great efficiency gain (Tables 5 and 6).

---

### Official Review · Reviewer_dQs2 · 2025-10-31

**Soundness:** 3
**Presentation:** 3
**Contribution:** 3
**Rating:** 6
**Confidence:** 3

**Summary:**

This paper introduces InComeS, a novel and flexible framework for efficient model editing in LLMs. The method is designed to overcome the scalability and efficiency limitations of traditional In-Context Learning (ICL) for editing, where performance degrades as the number of edits increases due to the finite context window. The core contribution of InComeS is a two-stage process of compression and selection. First, each piece of editing information is independently compressed into the KV cache of a special gist token. Second, the model is augmented with specialized cross-attention modules. These modules enable the model to dynamically and selectively attend to the pool of compressed gist tokens at inference time, retrieving the most relevant information for a given query. The authors conduct extensive experiments on a variety of complex model editing benchmarks. The results demonstrate that InComeS consistently outperforms a wide range of existing editing methods, showing marked improvements over the strong ICL baseline.

**Strengths:**

1. The integration of gist-based context compression with a learnable, dynamic selection mechanism is a novel combination that directly addresses the bottlenecks of ICL for batch editing.

2. By compressing edits in parallel and using a lightweight selection mechanism, it offers substantial speedups over ICL. This makes the approach practical for real-world applications.

3. This paper is clearly written, well organized, and generally easy to understand.

**Weaknesses:**

1. The method requires a continued pre-training phase to teach the model the compression and selection mechanisms.

2. The results in Table 1 show that the improvement of InComeS on Llama-3.2-1B is much greater than that on Qwen2.5-7B. This may indicate that the effectiveness of the method diminishes as the model scale increases.

**Questions:**

See weaknesses.

---

> ### Author Response · Authors · 2025-11-21
>
> Thank you very much for your valuable reviews. We respond to your concerns as follows:
>
> > 1. The method requires a continued pre-training phase to teach the model the compression and selection mechanisms.
>
> Unlike the traditional training paradigm adopted in SERAC[1], which requires training when data distribution changes, our method only needs an overall meta-training to enable the model to learn how to compress and select the relevant editing information. After that, no more training is needed, and the model is applicable to all data distributions. For example, the results of our method for different editing scenarios in Tables 1, 2, and 3 are from the same model trained from a single meta-training. Considering the great performance and efficiency gain (Tables 5 and 6), we believe this one-time meta-training cost is acceptable.
>
>
>
> > 2. The results in Table 1 show that the improvement of InComeS on Llama-3.2-1B is much greater than that on Qwen2.5-7B. This may indicate that the effectiveness of the method diminishes as the model scale increases.
>
> Thank you for this nice question. To further verify your point, we run an extra experiment for Llama-3.1-8B on the dataset MQuAKE. The results are as follows ("s" represents "single edits", "b" means "batch edits (100)"):
>
> |Method                 |2-edits-s |3-edits-s |4-edits-s |2-edits-b |3-edits-b  | 4-edits-b |
> | --------------------- | -------- | -------- | -------- | -------- | --------  | --------  |
> |ICL (Llama-3.2-1B)     | 0.5923   | 0.5900   | 0.5163   | 0.5006   | 0.4986    | 0.4237    |
> |InComeS (Llama-3.2-1B) | **0.7119** | **0.7217** | **0.7262** | **0.5393** | **0.5279**   | **0.5273**    |
> |ICL (Llama-3.1-8B)     | 0.5876   | 0.5699   | 0.4547   | 0.4864   | 0.4958    | 0.3951    |
> |InComeS (Llama-3.1-8B) | **0.7439**   | **0.7825**   | **0.7524**   | **0.5628**   | **0.5655**    | **0.5190**    |
>
> For the two Llama series models, the improvement does not diminish. This indicates that the observed improvement decreases are likely attributable to the models' specific characteristics developed during their training process, rather than their model size.
>
> Despite the performance variance, our method has a great efficiency gain (Tables 5 and 6) compared to other baselines, including ICL.
>
> [1] Memory-Based Model Editing at Scale

---

### Official Review · Reviewer_CWX5 · 2025-10-31

**Soundness:** 3
**Presentation:** 3
**Contribution:** 2
**Rating:** 4
**Confidence:** 3

**Summary:**

InComeS proposes a framework to edit the LLM through compression and selection mechanisms. The authors demonstrate that InComeS offers improved efficiency and accuracy compared to strong baselines and test on several challenging datasets and complex editing scenarios. The method's effectiveness is supported by results on multi-hop, natural language, and editing tasks which need reasoning ability, demonstrating strong scalability and adaptability across diverse editing tasks. And it is further analyzed through extensive ablations and analysis experiments.

**Strengths:**

1. The paper proposes the use of KV cache to improve efficiency and proposes corresponding training algorithms to improve the performance, and cross-attention modules are added to dynamically select the most relevant information gist.

2. The paper conducts a large number of experiments and thorough analysis.

**Weaknesses:**

1. There are only two models used in the paper, and the latest model (e.g. Qwen3-8B) is not used. If results about this model are reported, it will be more convincing. If the time is not sufficient, the author could consider only adding a small number of baselines for comparison.

2. The performance of the model in Table 3 is not competitive.

3. The paper uses the method of compressing content into key-value cache of gist tokens to achieve this. However, for different models, different gist token representations need to be saved. This method is similar to RAG, but the contents of RAG are visible. However, it is difficult to trace the contents of gist, which may cause some problems for understanding. And the content searched by RAG can be used by different model without other preprocessing. But the gist is only used for only one model. The design of the InComeS is not complete as RAG. For example, What should be done if the edited content is duplicated, and how to maintain all vectors in pool, like insert new gist and remove? The author did not design a special module to handle this situation.

**Questions:**

The ICL method in the paper is to fill all the information into the context. If the RAG method is used and only the most relevant information is selected for filling, what would be the result and efficiency?

---

> ### Author Response · Authors · 2025-11-21
>
> Thank you very much for the insightful review. We would like to respond to your concerns as follows:
>
> > 1. There are only two models used in the paper, and the latest model (e.g., Qwen3-8B) is not used. If results about this model are reported, it will be more convincing. If the time is not sufficient, the author could consider only adding a small number of baselines for comparison.
>
> Thank you very much for your advice. We select several baselines that have been demonstrated to be powerful in the existing results for the experiments. The results are as follows ("s" means "single editing", "b" means "batch editing"):
>
> MquAKE (Qwen3-8B):
> | Method   | 2-edits-s| 3-edits-s| 4-edits-s| 2-edits-b| 3-edits-b| 4-edits-b|
> | -------  | -------- | -------- | -------- | -------- | -------- | -------- |
> | base     | 43.49    | 41.01    | 26.32    | 43.49    | 41.01    | 26.32    |
> | FT-M     | 51.32    | 50.39    | 45.22    | 48.11    | 44.24    | 41.39    |
> | LoRA     | 38.95    | 36.27    | 33.02    | 24.41    | 21.24    | 22.12    |
> | MEMIT    | 49.14    | 46.05    | 41.15    | 42.34    | 40.10    | 33.45    |
> | ICL      | 55.82    | 54.13    | 47.06    | 50.78    | 48.84    | 45.28    |
> | InComeS  |**56.36** |**58.71** |**59.93** |**53.42** |**50.90** |**50.36** |
>
> DUNE (Qwen3-8B):
> | Method   | new info-s| scientific-s| de-biasing-s| new info-b| scientific-b| de-biasing-b|
> | -------  | --------  | --------    | --------    | --------  | --------    | --------    |
> | base     | 82.99     | 86.62       | 31.07       | 82.99     | 86.62       | 31.07       |
> | FT-M     | 78.23     | 75.44       | 56.38       | 76.33     | 72.34       | 54.28       |
> | LoRA     | 71.23     | 69.99       | 43.23       | 73.13     | 64.59       | 42.22       |
> | SERAC    | 81.29     | 80.45       | 49.65       | 80.99     | 78.53       | 45.66       |
> | ICL      | 83.06     | 81.12       | 45.45       | 83.35     | 82.76       | 30.34       |
> | InComeS  | **83.17** | **85.36**   | **63.96**   | **84.17** | **87.57**   | **65.96**   |
>
> The results show that our method continues to perform better than other baselines on the latest model.
>
>
>
> > 2. The performance of the model in Table 3 is not competitive.
>
> Indeed, the existing baselines show superiority on the **edit success** metric in Table 3, indicating they perform well in recalling exact edit facts (lines 013 - 014). However, we argue that this metric may not be a comprehensive metric to evaluate the real capability of the post-edited model, as it is vulnerable to **overfitting problem** [1], i.e., the model can assign disproportionately high probabilities to the edit target to get a high edit success rate. This may explain why the simple fine-tuning method dominates other baselines in this metric.
>
> Therefore, we focus on complex settings, including multi-hop editing, natural language editing, and portability that aligns more with the real application scenarios. Our method shows great performance on these settings, which aligns with our motivations (lines 035-037). Note that in the portability metric, the fine-tuning method no longer prevails like it does in the "edit success" metric, indicating that this metric cannot be "cheated" via overfitting. In addition, the results in Table 7 (complete version of Table 3 in Appendix) show that our method maintains a good **locality** performance compared to other baselines. Again, despite the high rate of edit success of fine-tuning, its locality collapses, indicating the existence of overfitting.
>
> [1] UNCOVERING OVERFITTING IN LARGE LANGUAGE MODEL EDITING

---

> ### Author Response · Authors · 2025-11-21
>
> > 3. The paper uses the method of compressing content into key-value cache of gist tokens to achieve this. However, for different models, different gist token representations need to be saved. This method is similar to RAG, but the contents of RAG are visible. However, it is difficult to trace the contents of gist, which may cause some problems for understanding. And the content searched by RAG can be used by different model without other preprocessing. But the gist is only used for only one model. The design of the InComeS is not complete as RAG. For example, What should be done if the edited content is duplicated, and how to maintain all vectors in pool, like insert new gist and remove? The author did not design a special module to handle this situation.
>
> Please notice that although we transform an edit sequence into gist representations, the mapping between each edit sequence and its corresponding gist representations is one-to-one. Given this mapping, we can manipulate the pool in the original text space as in the plain RAG setting.
>
> We respond to your concerns one by one.
> - "it is difficult to trace the contents of gist, which may cause some problems for understanding." -> We believe a mapping between the text and its gist representations could be built here to solve the problem. As the setting of our method strictly follows one edit - one gist (Figure 1), the mapping would not be hard to build.
> - "the content searched by RAG can be used by different model without other preprocessing. But the gist is only used for only one model." -> Our motivation is to design an ICL-like editing method that inherits the powerfulness and flexibility of ICL and alleviates its limitations in efficiency and retrieval accuracy when context grows. It targets the application scenario of **ICL** instead of RAG. Specifically, after the **one-time** meta training, our method can serve as an alternative to ICL. Therefore, our method is also a downstream approach, just like ICL, which can take the output from upstream.
> - "The design of the InComeS is not complete as RAG. For example, What should be done if the edited content is duplicated, and how to maintain all vectors in the pool, like insert new gist and remove?" -> As one edit strictly corresponds to one gist (mentioned in the first point), the mapping between text and gist can be constructed. Then we safely conduct the mentioned operations like deduplication, insertion, and removal in the text space.
>
>
>
> > Question: The ICL method in the paper is to fill all the information into the context. If the RAG method is used and only the most relevant information is selected for filling, what would be the result and efficiency?
>
> We compare our method and RAG in Appendix E.1, where we show the difference between them over problem settings, methodology, and the multi-hop editing problem. The results are shown in Table 8. The single editing results for RAG in this table are the same with ICL since it does not need any retrieval. For batch editing, RAG is allowed to retrieve the 10 most relevant documents from the batch. The results show it even fails to beat ICL. As for the efficiency, as mentioned above, our method targets downstream application scenarios like ICL. Therefore, the context information should remain the same for fair comparison.

---

### Note · Authors · 2026-01-03

I have read and agree with the venue's withdrawal policy on behalf of myself and my co-authors.